# Pharmacists’ Satisfaction with Work and Working Conditions in New Zealand—An Updated Survey and a Comparison to Canada

**DOI:** 10.3390/pharmacy11010021

**Published:** 2023-01-23

**Authors:** Sharon Jessie Lam, Larry D. Lynd, Carlo A. Marra

**Affiliations:** 1School of Pharmacy, University of Otago, Dunedin, Otago 9054, New Zealand; 2Faculty of Pharmaceutical Sciences, University of British Columbia, Vancouver, BC V6T 1Z4, Canada

**Keywords:** working conditions, satisfaction, mental health

## Abstract

Background: As roles have evolved over time, changes in workplace environments have created higher patient expectations creating stressful conditions for pharmacists. Aim: To evaluate pharmacists’ perceptions of their working conditions, work dissatisfaction, and psychological distress; determine their predictors in New Zealand (NZ); and compare results with Canadian studies and historic NZ data. Methods: A cross-sectional online survey was distributed to registered pharmacists in NZ. The survey included demographics, work satisfaction, psychological distress, and perceptions of their working conditions (six statements with agreement rated on a 5-point Likert scale). Comparisons were made with surveys from Canada and NZ. Chi-square, t-tests, and non-parametric statistics were used to make comparisons. Results: The response rate was 24.7% (694/2815) with 73.1% practicing in a community pharmacy (45.8% independent, 27.3% chains). Pharmacists disagreed on having adequate time for breaks and tasks, while the majority contemplated leaving the profession and/or not repeating their careers again if given the choice. Working longer hours and processing more prescriptions per day were predictive factors for poorer job satisfaction. More NZ pharmacists perceived their work environment to be conducive to safe and effective primary care (57% vs. 47%, *p* < 0.001) and reported that they had enough staff (45% vs. 32%, *p* = 0.002) as compared to Canadian pharmacists. Pharmacists’ job satisfaction and psychological distress have not improved compared to the assessment 20 years prior. Conclusions: NZ pharmacists perceive working conditions to be sub-optimal yet had higher satisfaction than their Canadian counterparts. Work dissatisfaction and psychological distress are high and have not improved over the last two decades.

## 1. Introduction

To accommodate pharmacists’ expanding scope of practice and services, changes to workplace environments are required [1,2]. However, in many jurisdictions, these changes have been slow to be implemented producing additional work-related stress [3,4]. Studies around the globe have continued to identify common factors contributing to workplace stress for pharmacists in the past decade [5,6,7,8,9,10,11,12,13,14,15,16,17,18,19,20,21,22,23,24].

In the United Kingdom and the United States, studies have found that increases in prescription volume and workload were positively correlated with an increase in pressure which can result in a lower standard of work and job satisfaction [5,6,7,8,9,10,11]. In addition, it also manifests as an increase in dispensing errors due to inadequate break time [12,13], workflow interruption [13,14], increased workload and stress [7,8,10,12,13,14,15,16], and dissatisfaction with the dispensary layout [6,10,12]. Insufficient staffing levels [8,15,16], a recent change in pharmacy ownership [5,17], and public expectations have all exhibited negative effects on pharmacists’ perception of work safety as well [13]. As a result, this pressure threatens pharmacists’ feelings of competency at work, despite the fact that their skills are being under-utilised due to a mismatch of education, role, and daily tasks [18,19,20].

Adapted from the Oregon Board of Pharmacy survey on pharmacists’ perception of their working conditions [21], Tsao et al. initially surveyed pharmacists working in British Columbia, Canada in 2013 [22] and then extended the survey to include four other Canadian provinces (Alberta, New Brunswick, Newfoundland, and Prince Edward Island) to further identify aspects that contributed to Canadian pharmacists’ perspectives of their working conditions [23]. The results show that the majority were satisfied that their working environment assisted in delivering safe and effective patient care; however, most were dissatisfied by the inadequate break time for lunch, staffing levels, and duration to complete their task at hand. Dissatisfaction towards their working conditions was more prominently found in pharmacists working in chain pharmacies, submerged by high prescription volume, and imposed by clinical service quotas for reimbursable tasks. These elements also contributed to perceived unsafe working conditions that can result in worse patient care.

New Zealand pharmacists share a similar scope of practice with Canadian pharmacists including providing vaccinations, medicines reviews, and treatment of minor ailments; although, the details and procedures for each service are slightly different. In addition, New Zealand pharmacists are remunerated similarly and face similar funding situations as Canadian pharmacists. As such, New Zealand pharmacists might be experiencing similar dissatisfactions as Canadian and US pharmacists. Nationally in New Zealand (NZ), a 2001 study piloted by Dowell et al. assessed the job satisfaction and psychological stress among NZ’s general practitioners (GP), physicians, surgeons, and community pharmacists [25]. Of the study participants, 41% of pharmacists exhibited significantly more work distress compared to 20–30% of GPs and surgeons. Pharmacists were also more likely to report that they were likely to seek other employment within the next year. This survey has not been repeated in NZ, and it is unclear if conditions have since improved.

Evaluations of pharmacists’ working conditions have been investigated by several countries globally, but not yet in New Zealand. As such, the objectives of this study are as follows:To determine the perception of current pharmacists’ working conditions in NZ and compare these with those recently found in Canada;To depict the work-related and psychological distress of NZ pharmacists and compare with a prior assessment;To explore relationships between work conditions and various predictor variables.

## 2. Method

### 2.1. Survey Administration/Mechanics

A cross-sectional online survey was created and hosted on the platform Qualtrics^©^ and distributed to currently registered pharmacists practicing in NZ via email with a web link. The Pharmacy Council of NZ (the regulatory authority) maintains a pharmacist database where members have previously consented to be contacted to participate in research and surveys. We utilised this resource to contact practicing pharmacists in NZ using their email addresses in the database and invited them to complete the survey anonymously. The survey design was adapted from the Oregon Board of Pharmacy survey [21]. The same survey was adopted by Tsao et al. as well to investigate working conditions for Canadian pharmacists. [22,23] Ethics approval for this study was granted by the University of Otago Human Ethics Committee on December 2020 (D20/418), and all participants provided informed consent.

### 2.2. Survey Instrument

The survey was comprised of pharmacists’ demographics, pharmacy experiences and clinical activities, characteristics of the pharmacy, perceptions of working conditions and work stress, and the General Health Questionnaire-12 (GHQ-12).

To assess perceptions of working conditions, the respondents were provided with 6 statements regarding their working conditions which were adapted from Tsao et al. [22]. Using a 5-point Likert scale (ranging from 1 = strongly disagree to 5 = strongly agree), respondents were asked to rate their agreement with the statements. All 6 statements were framed in such a way that an agreement with them was indicative of satisfaction with working conditions.

Nine statements regarding participants’ perceived work stress were provided to participants, who were subsequently asked to rate each statement on a 7-point Likert scale for agreement (1 = strongly disagree and 7 = strongly agree). These statements were adapted from Dowell et al. [25].

The GHQ-12 [26,27,28] is a validated and reliable instrument. The General Health Questionnaire (GHQ) is an instrument developed by Goldberg and Hillier in 1979 for screening psychological distress among adults working in primary care settings [1,26]. Goldberg et al. proceeded to generate a shorter but equally robust version known as the GHQ-12, which contains 12 items [27]. Respondents rank each statement on a relative scale of feeling “never” to “always” for each item. Following the responses, results are dichotomised to either 0 (none or less than usual) or 1 (more than usual) to evaluate anxiety and depressive symptoms in individuals lacking psychiatric disorders. The total possible score is 12 and a score of 4 and above is considered to be indicative of the possibility of suffering from potential psychiatric morbidity [25,26,27]. This questionnaire was included to determine the presence of psychiatric impacts on our respondents’ jobs and to also allow our data to be comparable to the 2001 study by Dowell et al. [25].

### 2.3. Data Collection

The survey was launched on 15 March 2021, and data were collected up to 20 May 2021. A follow-up email was sent to participants after one month of initially launching the survey and again after another month as reminders to increase responses. Participants’ consent was obtained prior to commencing the survey and the anonymity of participants was guaranteed.

### 2.4. Data Analyses

We used summary statistics to describe the demographics of respondents and results from the survey questions and the GHQ-12 questionnaires (Appendix A). The Likert scale assessing the working conditions statements was truncated to combine options “strongly agree” with “agree” while “strongly disagree” with “disagree”. Hereafter, the options will be referred to as ‘agree’ and ‘disagree’ respectively, permitting comparability with results from Tsao et al.’s studies [22,23]. We categorised our respondents into community, hospital, and other pharmacists, and we categorised the self-reported hours of work each week (<20 h; ≤20 and <40 h; ≤40 and <60 h; and >60 h), and the number of prescriptions filled per day (≤100, and >100). To analyse and compare results from the Likert scales, the non-parametric Mann–Whitney U test with post hoc pairwise comparisons was used to compare medians. T-tests and ANOVA were used to compare the mean scores of the GHQ-9 between our sub-groups. Chi-squared tests were used to determine the statistical significance between the percentage of pharmacists who scored above the cut-off scores of 4 and 8 in the GHQ-12 questionnaire in our study and Dowell et al.’s study from 2001 [25]. A *p*-value < 0.05 was considered statistically significant.

## 3. Results

### 3.1. Respondents

Of the 2815 surveys sent out to registered pharmacists in New Zealand, 694 pharmacists responded (response rate of 24.7%); however, only 579 were available for full analysis due to missing data. A detailed illustration of the respondents’ demographics is characterised in Table 1. These demographics are very similar to the Canadian respondent pharmacists that were assessed by Tsao et al. who form our comparator sample [22,23].

### 3.2. Personal Pharmacy Experiences and Clinical Activities

The majority (52.0%) of the respondents reported working an average of 8.1–10 h per shift, and 47.7% reported working more than 40 h in a typical week. In terms of filling >100 new prescriptions/day or >100 refills per day, the results were 42.2% and 36.0%, respectively.

### 3.3. Characteristics of the Pharmacy

A total of 81.5% of the respondents confirmed their primary practicing site offered a private consultation room. In total, 73.8% reported that the average patient wait time was no longer than 20 min. A total of 46.2% reported their site to be spacious while 20.9% felt it was neither spacious nor cramped, and the remaining 26.4% felt that their practice site was cramped.

### 3.4. Satisfaction with Working Conditions

In general, the majority of the respondents disagreed on having adequate time for breaks/lunches (46%) and were dissatisfied with the amount of time they had to complete their job (51%). However, most agreed that their primary practice site fostered an environment conducive to providing safe and effective primary care (57%) and also had adequate staffing levels, which included pharmacists, technicians, and retail assistant staff members (Table 2).

When comparing the median Likert scores for “I have adequate time for break/lunches at my practice site”, the median score of those who worked 40 h or less compared to those who worked more than 60 h was significantly higher (*p* = 0.047). Similarly, for the statement “I am satisfied with the amount of time I have to do my job”, those who worked less than 40 h or less a week had significantly higher scores (*p* = 0.004). Similar differences were revealed when analysing daily prescription volume categories. When comparing the median Likert scores of the agreement for “My site has adequate pharmacist staff to provide safe and effective primary care”, “My site has adequate technician staff to provide safe and effective primary care”, and “My site has adequate retail assistants to provide safe and effective primary care”, those with new prescription volumes ≤100 had higher median scores (*p* = 0.004, *p* = 0.024, and *p* = 0.028, respectively).

To compare NZ and Canadian pharmacists, data collected from Tsao et al.’s study and ours are placed adjacent to one another in Table 2 [23]. A similar distribution of primary practice roles was found for the two samples. Scores for “My employer provides a work environment that is conducive to providing safe and effective primary care” and “ My site has adequate technicians staff to provide safe and effective primary care” were significantly higher in NZ as compared to Canada.

Levels of perceived work stress and job dissatisfaction affecting NZ pharmacists are presented in Table 3. When compared to the 2001 findings, the 2021 survey results show that the level of perceived work stress has mostly remained stable in areas such as contemplating leaving their job because of work stress, the feeling that work stress has affected health, and, if given the choice, they would not choose pharmacy as a career again.

The results of the GHQ-12 scores are displayed in Table 4. From 2001 to 2021, the mean GHQ-12 score for pharmacists in NZ has remained relatively stable (3.21 to 3.50, *p* = 0.13) and the percentage of those scoring four and above has also remained stable (40.7% to 47%, *p* = 0.11). However, those who scored higher than eight decreased from 11.3% to 2% (*p* < 0.001).

## 4. Discussion

This is the first study to evaluate the working conditions of registered pharmacists in NZ which allows for a comparison with similar work conducted in Canada. It also re-assesses their satisfaction toward their job and the impact of their workload on their well-being and facilitates a comparison with a similar survey conducted 20 years prior. Our study determined that perceived working conditions for New Zealand pharmacists were substandard and were similar to those perceived by Canadian pharmacists. We discovered that, although more than half acknowledged their workplace was conducive to delivering safe and effective patient care, numerous pharmacists were troubled by not having sufficient time for their breaks and for their job. A large proportion of pharmacists had considered abandoning their pharmacy career due to the work impact on their health, and most would be unlikely to repeat their career choice again if given the option. Even more were irked by the bureaucratic micromanagement and the incessant amount of administrative paperwork they have to involuntarily wrestle as part of their jobs. We found a high degree of psychological distress in pharmacists that was higher than it was 20 years ago.

Findings in this research generally resonated with those conducted by others internationally. Using the same set of descriptive statements to describe their working environment allowed our results to be compared with results gathered from Tsao et al.’s studies [22,23] and the Oregon Board of Pharmacy Study [21]. In general, our findings were similar to these other similar surveys with a low endorsement for “I have adequate time for break/lunches at my practice site”, “I am satisfied with the amount of time I have to do my job”, “My site has adequate pharmacist staff to provide safe and effective primary care” and “my site has adequate retail assistants to provide safe and effective primary care”. There was significantly more agreement among NZ pharmacists than among Canadian pharmacists for “My employer provides a work environment that is conducive to providing safe and effective primary care” and “My site has adequate technician staff to provide safe and effective primary care”. Some of the findings from the Canadian survey associated quotas imposed by employers for paid clinical services with a lower agreement with the working condition statements. In NZ, in general, there are fewer paid services from the government for patient care services based in pharmacies than in Canada, thus making the imposition of quotas on pharmacists less common which may explain some of the differences. With a lower time imposition on pharmacists to provide these services, there may be less reliance on technicians to make the dispensary operations run, thus reflecting the higher satisfaction with the number of technicians in the NZ system. More research is needed to elucidate the reasons behind these differences.

When comparing the survey results from the current survey to the questions that were asked 20 years prior, there was little improvement in pharmacist satisfaction and mental health. Generally speaking, the majority of pharmacists felt work stress has compromised their health, family, and social life. They felt submerged by paperwork and frustrated by bureaucratic interference. Most agreed that government funding to support pharmacists in delivering patient care has increased minimally over the years and that pharmacists should most definitely play a greater role in primary care. With a larger sample size but a heightened psychological distress score, it highlights the concerns recognised and addressed 20 years ago that have not only been sustained but have also regressed for community pharmacists.

### 4.1. Strengths

By conducting a cross-sectional study design, it allows for the identification of potential public health issues for current New Zealand pharmacists, which is difficult to address using other study designs. Recognition and acknowledgement of the current working conditions for New Zealand pharmacists are important as it serves as a driving force for improvement and progression for this group in the workforce. Raising awareness of the troubles pharmacists face in New Zealand provides a foundation and stepping-stone to plan and implement actions required for addressing areas requiring improvements. This can foster an upgrade in working environments for New Zealand pharmacists leading to improvements in mental and physical distresses and satisfaction in the future.

This study also incorporated a reasonably large sample size. Based on the New Zealand Pharmacy Council Working Demographics 2020 [29], there are a total of 3906 practicing pharmacists in New Zealand. Of the 3906 practicing pharmacists, 2815 have consented to enrol in the database that allowed researchers to invite them as participants for research and surveys. Having a relatively large sample size provides an opportunity for more accurate and representative results, which allows findings from this study to be applicable to a greater population.

### 4.2. Limitations

Despite the strengths, there are also limitations present that should be acknowledged. Firstly, the response rate was modest despite the relatively large sample size, which indicates the presence of a non-response bias. This may be driven by a limited time frame of two months for data collection. However, in NZ [29], 66.6% of practicing pharmacists were females and 79% worked in a community pharmacy which is similar to our sample of 65.9% females and 73.1% community pharmacists. The median age reported for NZ registered pharmacists in 2020 was 37.6 years old while the mean age of our responders was 44.9 years old. As such, our sample is congruent with the pharmacist population, suggesting that the results may be generalisable to the wider population.

Secondly, based upon the results of our initial piloting of the survey, the wording of the survey was updated to more closely reflect terms that were reflective of New Zealand pharmacy practices. Consequently, in the working conditions section, the item “patient care” was replaced with “primary care” which could have led to a slightly different interpretation of these questions. However, we believe the risk of this was low as those pharmacists who piloted the survey suggested the new wording based on the original interpretation.

Additionally, due to the cross-sectional study design, the causality of items (work hours, prescription volume) on job stress and working conditions cannot be determined. Despite evidence that the participants were stressed and dissatisfied, this could potentially be influenced by non-work-related factors. However, comments in the free text section on the reasons for their increasing stress and dissatisfaction levels were about being underpaid, understaffed, and undervalued while simultaneously being overworked. However, we recognised that not all participants commented on the open-ended question; thus, such comments may not be representative of all participants, nor is it generalisable to the national population. Additionally, ideally, there would have been more recently published evaluations of pharmacists’ working conditions and work stress than that conducted by Dowell et al. [25] to form a comparison. Unfortunately, these were the most recent data available in New Zealand, highlighting the need for regular evaluations to occur.

Finally, response bias may be present where participants respond to survey questions by attempting to predict the researchers’ desired results. Attempts to minimise response bias have been carried out by promising anonymity to all participants; however, subjectivity cannot be eliminated completely. Although conducted during 2021, a period when the COVID-19 pandemic continued to plague the globe, we must view the responses in this context. However, the perceptions of the surveyed pharmacists are likely not entirely due to COVID-19 as the survey was also conducted a year after the pandemic emerged, which suggests adaptations in response to the outbreak have already been well-established. Furthermore, many additional comments left by participants signified that increases in stress levels and unsatisfactory job conditions have been a long-standing issue plaguing New Zealand pharmacists.

## 5. Conclusions

This study revealed that NZ pharmacists’ perception of their current working conditions is suboptimal. Although similar to Canadian pharmacists’ perceptions, NZ pharmacists had more agreement with the statements that employers provide environments that are conducive to safe primary care practice and that there was enough technical staff. Pharmacists, especially those who worked longer hours and processed more prescriptions per day, strongly agree with statements of contemplating leaving their job due to stress and the likelihood of not repeating the same career. Psychological distress among pharmacists remains high and remained largely unchanged over the past 20 years. Further evaluations conducted more regularly can monitor working conditions, investigate solutions for changes to foster a better workplace environment for pharmacists, and characterise the impact of pharmacists’ working conditions on patient-centered care.

## Figures and Tables

**Table 1 pharmacy-11-00021-t001:** Respondent demographics.

Characteristics	Number (n = 579)	Percentage (%)
**Gender** Female Male	382 197	65.9 34.1
**Qualifications** Bachelor of Pharmacy (BPharm) Diploma of Pharmacy Postgraduate qualifications (e.g., PGCertPharm, PGCertPharPres, etc.) Others (e.g., PhD, BSc, PGDip Management, etc.)	294 93 138 54	50.8 16 23.8 9.4
**Primary Practice** Site Community pharmacy (chain or independent) In-patient hospital pharmacy Others (e.g., academia, primary care, industry, etc.)	423 84 72	73.1 14.4 12.5
**Primary Practice Site Location** Main urban area (≥30,000 people) Secondary urban area (10,000~29,999 people) Minor urban/rural area (<9999 people)	362 122 95	62.5 21.0 16.5
**Practice Role** Clinical/specialist pharmacist Pharmacy manager Pharmacy director/owner Staff pharmacist Others (e.g., industry, academia, etc.)	103 88 126 179 83	17.8 15.2 21.8 30.9 14.3
**Years as Registered Pharmacist** ≤5 6–15 16–24 ≥25	106 153 92 219	18.3 26.4 15.8 37.9

**Table 2 pharmacy-11-00021-t002:** Responses from NZ pharmacists’ assessment of working conditions as compared with Tsao et al.’s 2016 Canadian study results [22].

	Canadian Pharmacists n = 1016		NZ Pharmacists n = 579	
	Disagree (%)	Agree (%)	Disagree (%)	Agree (%)
“I have adequate time for break/lunches at my practice site”	48	40	46	40
“I am satisfied with the amount of time I have to do my job”	45	34	51	31
“My employer provides a work environment that is conducive to providing safe and effective primary care” ^	28	47	20	57
“My site has adequate pharmacist staff to provide safe and effective primary care”	36	41	32	46
“My site has adequate technician staff to provide safe and effective primary care” *	35	32	33	45
“My site has adequate retail assistant staff to provide safe and effective primary care”	31	45	25	42

^ Results are statistically significant with *p* < 0.001; * Results are statistically significant with *p* = 0.002 as tested with Chi-square tests. The “neutral ratings were not included, and thus, the agree/disagree proportions do not add to 100%.

**Table 3 pharmacy-11-00021-t003:** Comparison of community pharmacists’ responses to statements about work stress with Dowell et al.’s 2001 study results [25].

	2001 NZ Pharmacists n = 303	2021 NZ Pharmacists n = 423	2021 NZ Pharmacists (Proportion Agree/Strongly Agree)
Contemplated leaving your job due to work stress	4 (2–6)	5 (3–7)	45%
Felt that work stress has affected health	4 (3–5)	5 (3–7)	45%
Felt unable to remain competent at work	3 (2–4)	3 (1–5)	15%
Felt overwhelmed by paper work	6 (5–7)	5 (3–7)	40%
Felt frustrated over bureaucratic interference	7 (6–7)	6 (4–7)	58%
Feel work has interfered significantly with family/social life	5 (4–6)	5 (3–7)	38%
Likelihood of not repeating career choice again	5 (3–6)	6 (4–7)	55%
Government funding for patient care has increased	2 (2–4)	1 (1–3)	3%
Pharmacists should have a greater role in primary care	5 (4–7)	6 (5–7)	71%

Results are shown in median (quartile range) with items scored on a 7-point Likert scale with 1 being strongly disagree and 7 being strongly agree. Additionally, results for the 2021 survey include the proportion that agreed and strongly agreed (scored as 6 and 7 on the Likert scale—these results were not available from the Dowell study). Of note, the proportion of those who disagreed/strongly disagreed with “Felt unable to remain competent at work”, and “Government funding for patient care has increased” were 47% and 72%, respectively.

**Table 4 pharmacy-11-00021-t004:** Comparison of community pharmacists’ General Health Questionnaire 12 (GHQ-12) scores with Dowell et al.’s 2001 study results [25].

	2001 NZ Pharmacists	2021 NZ Pharmacists
Count	n = 300	n = 389
Total GHQ-12 mean (SD)	3.21 (3.20)	3.50 (1.83)
Proportion GHQ score 4–12	41%	47%
Proportion GHQ score 8–12 *	11%	2%

* *p* < 0.001.

## Data Availability

Not available.

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
