# Peer review of "Pharmacists’ Satisfaction with Work and Working Conditions in New Zealand—An Updated Survey and a Comparison to Canada"

_pharmacy, 2023, doi:10.3390/pharmacy11010021_

Round 1

Reviewer 1 Report

This is a useful contribution to the ongoing discussions about pharmacy practice working conditions across the globe. 24.7% is a very good response rate, congratulations!

My comments generally relate to a lack of attention to detail in the writing and presentation, which has led to some errors.

Introduction

Line 42. Tsao et al published the article in 2016 but they collected the data in 2016. Tsao et al then went on to survey the rest of Canada in 2016 and published in 2020. Please check the accuracy of your statements.

Methods.

The first table cited in the text is Table 2. Tables and Figures should be cited in the order they are presented, so if you want me to look at Table 2 first, it is actually Table 1. I suggest that you don't actually need to refer to Table 2 at this point, as the statements are those of Tsao et al 2016 - so in fact that reference should be given in the methods instead of Table 2. However, as I now compare the statements, I see a difference between those in Tsao et al and yours - statements 4, 5 and 6 are 'patient care' in Tsao but 'primary care' in yours. Is this a mistake in the writing or a mistake in the survey? If the question was asked differently in the survey it will need to be addressed in the discussion, as the interpretation of the question could be different due to that one word.

The nine statements regarding perceived workplace stress (note it is workplace stress at line 87 but work stress at line 93 and elsewhere - please be consistent throughout the document - is it about the place or the work itself) - again where do these come from? Give the citation. The list of statements could be within the body of the text, perhaps as dot points, as the 8 points are currently presented as a table but don't have a Table number or legend, and probably don't really warrant a table anyway. But if these statements are all from Tsao and are repeated later in Table 3 then there is no need to write them out twice, just refer to the origin.

Table 1 could be formatted better so that it sits on a single page, e.g. separate columns for characteristics / n / %, put the characteristics text to the left (to make it easier to read) and remove the shading. The current layout matches that in Tsao et al 2020, but that's not the same layout as used in this MDPI journal.

Table 3 - check the wording of statement 8 because it doesn't make sense. Also statement 9, should that be Pharmacists plural?

Table 4 legend - explain what GHQ is and that a score above 4 is considered to indicate 'psychiatric morbidity'. (I took that from your methods - perhaps it should be 4 and above since you highlight 4-12 as a separate category in Table 4. I haven't accessed a reference to check.) At line 189-192, those p values are not significant, so it is more appropriate to say that the GHQ-12 score is similar between years rather than saying it has increased. The only change is a decrease in the very high scores.

Discussion is fine.

References - ref 21 and 29 author needs to be corrected.

Author Response

Line 42. Tsao et al published the article in 2016 but they collected the data in 2016. Tsao et al then went on to survey the rest of Canada in 2016 and published in 2020. Please check the accuracy of your statements.

We thank the reviewer for this comment. In the first Tsao et al. paper (CPJ 2016), the responses were collected from pharmacists practicing in British Columbia from October to November 2013.  In the 2nd Tsao et al., the data were collected in 2016 and published in 2020.  We have corrected the manuscript to reflect this more accurately.

Methods.

The first table cited in the text is Table 2. Tables and Figures should be cited in the order they are presented, so if you want me to look at Table 2 first, it is actually Table 1. I suggest that you don't actually need to refer to Table 2 at this point, as the statements are those of Tsao et al 2016 - so in fact that reference should be given in the methods instead of Table 2.

The reviewer is correct. The reference to Table 2 has been removed  and Tsao et al. from 2016 has been referenced instead.

However, as I now compare the statements, I see a difference between those in Tsao et al and yours - statements 4, 5 and 6 are 'patient care' in Tsao but 'primary care' in yours. Is this a mistake in the writing or a mistake in the survey? If the question was asked differently in the survey it will need to be addressed in the discussion, as the interpretation of the question could be different due to that one word.

The wording was updated after piloting the survey amongst New Zealand pharmacists.  We have added this change in wording as a discussion point in the limitations section.

The nine statements regarding perceived workplace stress (note it is workplace stress at line 87 but work stress at line 93 and elsewhere - please be consistent throughout the document - is it about the place or the work itself) - again where do these come from? Give the citation. The list of statements could be within the body of the text, perhaps as dot points, as the 8 points are currently presented as a table but don't have a Table number or legend, and probably don't really warrant a table anyway. But if these statements are all from Tsao and are repeated later in Table 3 then there is no need to write them out twice, just refer to the origin.

We thank the reviewer for their comments and have modified the text to be more congruent with our intended meaning. We have deleted the 8 points presented as a table and have referenced Dowell et al. as our source.

Table 1 could be formatted better so that it sits on a single page, e.g. separate columns for characteristics / n / %, put the characteristics text to the left (to make it easier to read) and remove the shading. The current layout matches that in Tsao et al 2020, but that's not the same layout as used in this MDPI journal.

We have reformatted Table 1 as per the reviewer's suggestions.

Table 3 - check the wording of statement 8 because it doesn't make sense. Also statement 9, should that be Pharmacists plural?

Agreed. Thanks for spotting this. We have corrected both of these statements.

Table 4 legend - explain what GHQ is and that a score above 4 is considered to indicate 'psychiatric morbidity'. (I took that from your methods - perhaps it should be 4 and above since you highlight 4-12 as a separate category in Table 4. I haven't accessed a reference to check.) At line 189-192, those p values are not significant, so it is more appropriate to say that the GHQ-12 score is similar between years rather than saying it has increased. The only change is a decrease in the very high scores.

The reviewer is correct in that it should be 4 and above and that the scores are not significantly different. The text has been changed to reflect this and the reviewer's comments.

Discussion is fine.

Thank you.

References - ref 21 and 29 author needs to be corrected.

These have been corrected - thanks for pointing out.

Reviewer 2 Report

The study evaluates Pharmacists' Perceptions of Their Working Conditions in New Zealand as Compared to Canada.

The topic is relevant and timely, Specifically, it is good to assess pharmacists' job satisfaction because there is a high demand for pharmacists and thus a need to ensure working pharmacists are retained.

The paper describes a basic survey with standard statistics.

Response rates were adequate for the analysis. Reporting of the results followed standard inferential analysis methods.

Methods:

Table 1. It is more readable to have a column labeled "n" and beside it one labelled "%" and then just put the numbers in each column.

Regarding a comparison of the Dowell et al.’s 2001 to the current 2021 study, it the Dowell data really the most recent available? Generally, we like to see more recent data for comparison.

Although, as you noted the results of the two studies were comparable, which is very discouraging as one would hope for improvement in work conditions over that length of time. At a minimum, you can add the lack of more recent data for comparison as a study limitation.

Conclusions: Please add some more suggestions for future work.

Author Response

The study evaluates Pharmacists' Perceptions of Their Working Conditions in New Zealand as Compared to Canada.

The topic is relevant and timely, Specifically, it is good to assess pharmacists' job satisfaction because there is a high demand for pharmacists and thus a need to ensure working pharmacists are retained.

Thank you for your comments.

The paper describes a basic survey with standard statistics.

Response rates were adequate for the analysis. Reporting of the results followed standard inferential analysis methods.

Thank you.

Methods:

Table 1. It is more readable to have a column labeled "n" and beside it one labelled "%" and then just put the numbers in each column.

We have reformatted Table 1 to comply with this request (and reviewer #1's request) .

Regarding a comparison of the Dowell et al.’s 2001 to the current 2021 study, it the Dowell data really the most recent available? Generally, we like to see more recent data for comparison.

Unfortunately, Dowell's study is the most recent data available which makes it even more imperative to conduct and publish a study looking at working conditions and work stress in pharmacists in New Zealand.

Although, as you noted the results of the two studies were comparable, which is very discouraging as one would hope for improvement in work conditions over that length of time. At a minimum, you can add the lack of more recent data for comparison as a study limitation.

We have added the lack of more recent data for comparison as a study limitation.

Conclusions: Please add some more suggestions for future work.

We have added suggestions for future work.

Reviewer 3 Report

General Comments

This research was a combination of 1) repeating a survey in NZ (published in 2001, cite 25) about work stress perceptions and 2) adapting/repeating a Canadian survey (published in 2016 and 2020, cites 22 and 23) about perceptions of work conditions (that was derived from a survey by the Oregon Board of Pharmacy).  Unfortunately, the authors have not written their description of the work this clearly and accurately. They describe (in the methods, lines 79-81) the survey as being adapted from the Oregon study and the “same” as the Canadian one, even though their current effort was this twofold effort.

This shortcoming in describing the survey content and design, as derived from two distinct surveys, is an example of work needed throughout the paper to ensure technical accuracy in writing and exposition for this work. There also are challenges with analyzing the data from perception scales and how to interpret the results.  In many places in the manuscript, language and expressions need careful review to ensure what is being written is interpreted and stated correctly.  In light of this, does the title only captures half of this research effort (and maybe it suggests two distinct papers might better serve the authors?).

Introduction

It may be prudent to simplify the introduction so that it more quickly gives a set-up for their research. For example, I have concern that the “conclusions” made in the first two sentences actually can be made from the citations noted.

Here is a suggestion for an alternate introduction section.  Start with a general statement that many studies have been  done to explore pharmacists perceptions of their work conditions as well as how work conditions can influence pharmacists’ satisfaction, stress, and subsequent performance and quality of care. In NZ, there have not been many studies, but repeating on that was done in 2001 might be merited and with similarities between NZ and Canada, a study there about work conditions could be adapted.  This kind of a more direct path to what your research is about may remove extraneous content and help ensure your set-up for the current research is targeted and accurate.

I think there really are only two objectives. First, determine perceptions of work conditions and compare with those recently found in Canada. Second, depict stress and compare with a prior assessment. (These combine 1, 2, and 4 as stated.)  Third, you explored selected relationships between some work conditions and situations and job satisfaction.

Methods

Study Design.  (Maybe Survey Administration/Mechanics is a better heading? If so, you can combine aspects of administration as included in this section along with the “data collection” aspects of contacting subjects, reminders, and how long the survey was open/available.) Qualtrics is a platform for conducting online surveys (you don’t “create” a survey with it). Your survey included items from the prior NZ and the Canadian studies (survey instrument description).  You were assisted in distributing the survey by the Pharmacy Council; they sent a message on your behalf with a link to the survey (I’m thinking this is how you did it since it is matches how we have used our licensing bodies to help us make contact and keep respondents anonymous, etc.).

Survey Instrument.  It appears from the appendix that the Likert ratings were not individually anchored with labels but only the “strongly” end points were anchored. Given that, it is curious that the working conditions used a 5-point scaling and the stress used a 7-point scaling. Was there a rationale for that difference in number of rating choices(more than merely duplicating the previous efforts)?  And, although I appreciate holding to the methods from prior work for comparison, can there really be a “neutral” middle rating on an agree/disagree scale?  Since the analyses were focused on agree/disagree, it may have been beneficial from the start to avoid this “awkward” middle rating.

Data Collection. When actually were the two reminders sent? Were they each one month after the initial email?  Or, were they sent after a couple weeks each, with open time for completing the survey up until the 20 May closing date? This needs to be clarified.

Data Analysis.  By truncating the responses to agree/disagree bimodal categories, the authors may be “throwing away” data.  If some of the items have higher rates of extreme ratings, that aspect of a descriptive data analysis could be informative. Similarly, since the “middle” (neutral?) ratings were not used for analyses, if/when more such responses occurred for specific survey items might be highlighted.  The analysis could include more complete “raw” descriptive data reporting, with the truncating for analyses that compared previous findings or explored relationships.  I also have concern using traditional parametric statistics (t-test, ANOVA) for categorical data for appropriateness or at least challenges in interpreting the results.  Care in explaining and describing methods (and results) is needed.

Results

Could more descriptive results be included? Maybe a table of the other responses than those highlighted? E.g., how many worked the different categories of hours? How can the median hours worked be 46.21 hours when this was a categorical variable? (Am I missing something?) The workload results are inaccurate; the quantification was persons with new or refill prescriptions, not numbers of prescriptions filled. The results of the workspace characterization do not add to 100%.  What was the missing proportion? (Incomplete responses?)

Table 2 might benefit from a footnote about how “neutral” ratings were not included and the agree/disagree proportions do not add to 100%. In some cases, around a third of responses were “neutral”; should that be mentioned? Why did the “staffing” items have so many that did not either agree or disagree?  Also, footnote what statistical test was performed and the difference was between the survey iterations.

Can what it means for a “median Likert score” be clarified?  Or, would there be benefit from calculating a mean score from the ratings (with the 1 through 5 or 7) calculated across respondents? Doing so can show via a simple number for comparing different ratings across items and surveys.

In Table 3, could an average score across respondents’ ratings be as useful for showing where the higher and lower ratings levels occurred? Using the 1 through 7 as values and computing averages would allow quick observation of how different from “neutral” (4.0) different items and different survey iterations were. It may be easier for readers to interpret the results. Addition insights could be added in text for higher proportions of “extreme” (strongly) agree/disagree ratings if those were present and more influential on the average scores.

Discussion

Care is needed to ensure the results are expressed accurately.  “Most” pharmacists would be represented by > 50% (and probably substantially more) providing a certain rating but it is not clear from the results presented in Table 3 that this occurred.  C/should these supporting results be included in the results (or perhaps added to Table 3 as a percent agreeing)?

Author Response

General Comments

This research was a combination of 1) repeating a survey in NZ (published in 2001, cite 25) about work stress perceptions and 2) adapting/repeating a Canadian survey (published in 2016 and 2020, cites 22 and 23) about perceptions of work conditions (that was derived from a survey by the Oregon Board of Pharmacy). Unfortunately, the authors have not written their description of the work this clearly and accurately. They describe (in the methods, lines 79-81) the survey as being adapted from the Oregon study and the “same” as the Canadian one, even though their current effort was this twofold effort.

Thanks very much for your comment. The other two reviewers did not have issue with the introduction and we are not sure what change you are asking us to make.  If the Editor believes that the Introduction needs to be changed, we are happy to comply.

This shortcoming in describing the survey content and design, as derived from two distinct surveys, is an example of work needed throughout the paper to ensure technical accuracy in writing and exposition for this work. There also are challenges with analyzing the data from perception scales and how to interpret the results. In many places in the manuscript, language and expressions need careful review to ensure what is being written is interpreted and stated correctly. In light of this, does the title only captures half of this research effort (and maybe it suggests two distinct papers might better serve the authors?).

Thanks very much for your comment. What you are asking, we believe, is whether the title captures the entire gist of our study and you believe that it only captures half of it.  We do agree that there is more to the study than represented by the title. As such, we propose:

Pharmacists' satisfaction with work and working conditions in New Zealand. An updated survey and a comparison to Canada.

Introduction

It may be prudent to simplify the introduction so that it more quickly gives a set-up for their research. For example, I have concern that the “conclusions” made in the first two sentences actually can be made from the citations noted.

We believe that the introduction is not overly long or inadequate. As such, we would prefer to keep it. However, if the Editor believes that it should be shortened and/or changed, we are happy to comply.

Here is a suggestion for an alternate introduction section. Start with a general statement that many studies have been done to explore pharmacists perceptions of their work conditions as well as how work conditions can influence pharmacists’ satisfaction, stress, and subsequent performance and quality of care. In NZ, there have not been many studies, but repeating on that was done in 2001 might be merited and with similarities between NZ and Canada, a study there about work conditions could be adapted. This kind of a more direct path to what your research is about may remove extraneous content and help ensure your set-up for the current research is targeted and accurate.

As above.

I think there really are only two objectives. First, determine perceptions of work conditions and compare with those recently found in Canada. Second, depict stress and compare with a prior assessment. (These combine 1, 2, and 4 as stated.) Third, you explored selected relationships between some work conditions and situations and job satisfaction.

We have combined the study objectives as the reviewer has suggested.

Methods

Study Design. (Maybe Survey Administration/Mechanics is a better heading? If so, you can combine aspects of administration as included in this section along with the “data collection” aspects of contacting subjects, reminders, and how long the survey was open/available.) Qualtrics is a platform for conducting online surveys (you don’t “create” a survey with it). Your survey included items from the prior NZ and the Canadian studies (survey instrument description). You were assisted in distributing the survey by the Pharmacy Council; they sent a message on your behalf with a link to the survey (I’m thinking this is how you did it since it is matches how we have used our licensing bodies to help us make contact and keep respondents anonymous, etc.).

We thank the reviewer for their comments. We have changed some of the explanation where the reviewers comments were accurate (ie. hosting of the survey on Qualtrics rather than creating it) but have kept our original text where it was accurate (mechanism by which we disseminated the survey).

Survey Instrument. It appears from the appendix that the Likert ratings were not individually anchored with labels but only the “strongly” end points were anchored. Given that, it is curious that the working conditions used a 5-point scaling and the stress used a 7-point scaling. Was there a rationale for that difference in number of rating choices(more than merely duplicating the previous efforts)? And, although I appreciate holding to the methods from prior work for comparison, can there really be a “neutral” middle rating on an agree/disagree scale? Since the analyses were focused on agree/disagree, it may have been beneficial from the start to avoid this “awkward” middle rating.

We thank the reviewer for their comments but the working conditions questions were not only anchored on both ends but each of the 5-points had a descriptor (see Q22 in the appendix).  There are neutral ratings on a Likert scale (the middle one is labelled "neither agree or disagree".  For comparison purposes (which is what we set out to do), we analysed the data from the end groups (agree and disagree) using Chi-square tests. We also believed that these were the most explanatory when it came to assessing the pharmacists' assessment of their working conditions.

Data Collection. When actually were the two reminders sent? Were they each one month after the initial email? Or, were they sent after a couple weeks each, with open time for completing the survey up until the 20 May closing date? This needs to be clarified.

We have clarified this further in the Data Collection section.

Data Analysis. By truncating the responses to agree/disagree bimodal categories, the authors may be “throwing away” data. If some of the items have higher rates of extreme ratings, that aspect of a descriptive data analysis could be informative. Similarly, since the “middle” (neutral?) ratings were not used for analyses, if/when more such responses occurred for specific survey items might be highlighted. The analysis could include more complete “raw” descriptive data reporting, with the truncating for analyses that compared previous findings or explored relationships. I also have concern using traditional parametric statistics (t-test, ANOVA) for categorical data for appropriateness or at least challenges in interpreting the results. Care in explaining and describing methods (and results) is needed.

We thank the reviewer for their comments. We believe that the neutral ratings from the rating scales (especially on the working conditions questions) are easy to determine.  We analysed the surveys the way that we did to allow comparison with the previous results which were part of our objectives.  We have added more "raw" results to the Appendix (attached) that can be published if the Editor recommends.  We followed advice from the following excellent article:  Harpe SE. How to analyze Likert and other rating scale data.  Currents in Pharmacy Teaching and Learning 2015;7:836-50.

Results

Could more descriptive results be included? Maybe a table of the other responses than those highlighted? E.g., how many worked the different categories of hours? How can the median hours worked be 46.21 hours when this was a categorical variable? (Am I missing something?) The workload results are inaccurate; the quantification was persons with new or refill prescriptions, not numbers of prescriptions filled. The results of the workspace characterization do not add to 100%. What was the missing proportion? (Incomplete responses?)

We have included the descriptive information that we want to include in this paper in Table 1 - we plan on publishing an additional paper where the remainder of the results will be explored. We agree that the mean number of weekly hours worked is not supported by the categorical nature of the data and we have deleted this statement.  All the numbers in Table 1 add to 100% now - previously there were some rounding issues and a typo that have been rectified.

Table 2 might benefit from a footnote about how “neutral” ratings were not included and the agree/disagree proportions do not add to 100%. In some cases, around a third of responses were “neutral”; should that be mentioned? Why did the “staffing” items have so many that did not either agree or disagree? Also, footnote what statistical test was performed and the difference was between the survey iterations.

We have added the footnotes to the Table as suggested by the reviewer.  We do not know why in the "staffing" questions, there seemed to be more "neutral responses" - from purely conjecture, it is possible that pharmacists were stepping up to the get the job done in the face of low staffing and thus rated this neutral while at the same time as experiencing more job stress (not enough breaks or time to do job the way that they would like).

Can what it means for a “median Likert score” be clarified? Or, would there be benefit from calculating a mean score from the ratings (with the 1 through 5 or 7) calculated across respondents? Doing so can show via a simple number for comparing different ratings across items and surveys.

We thank the reviewer for this comment.  However, we are not sure what the reviewer would like clarified.  Most review/opinion articles on the analysis of Likert scales conclude that the most appropriate measure of central tendency is the median - this is true especially when there is clustering around both ends of the scale as the mean could reside in the "neutral" space which does not adequately characterise the data (ref: Sullivan GM, Artino AR. Analyzing and interpreting data from Likert-type scales. J Grad Med Educ 2013;5(4):541-2.)

In Table 3, could an average score across respondents’ ratings be as useful for showing where the higher and lower ratings levels occurred? Using the 1 through 7 as values and computing averages would allow quick observation of how different from “neutral” (4.0) different items and different survey iterations were. It may be easier for readers to interpret the results. Addition insights could be added in text for higher proportions of “extreme” (strongly) agree/disagree ratings if those were present and more influential on the average scores.

We thank the reviewer for this comment. We do not have access to the 2001 data and these results were only presented in the format that is in Table 3. As such, we chose to present the results from the 2021 survey in a similar fashion to allow comparison across the time period.  We do agree that had we had the data from 2001, breaking these result out further would be beneficial, but without the 2001 data, we can only highlight the data from 2021. As such, we have added a column to Table 3 to highlight the proportion who agree/disagree with the statements.

Discussion

Care is needed to ensure the results are expressed accurately. “Most” pharmacists would be represented by > 50% (and probably substantially more) providing a certain rating but it is not clear from the results presented in Table 3 that this occurred. C/should these supporting results be included in the results (or perhaps added to Table 3 as a percent agreeing)?

We thank the reviewer for this comment and have changed the "most" wording. We have also added a column to Table 3 to add the results requested

Round 2

Reviewer 3 Report

This revision is an improved version.  There are some places that may benefit from additional attention to improve clarity and accuracy.

The revised title did not appear with this version.

A bit more clarity and explanation on survey administration could benefit others considering this kind of research. Did the NZ Pharmacy Council provide you the email addresses?  Or, did they assist by sending an email on your behalf with an invitation and link to the survey?  Was the consent based on the general agreement with the Council or did consent specific to this survey occur (as suggested later).

In the methods, adding the specific labels used for the Likert scaling (e.g., strongly agree, agree, mildly agree, and conversely for disagree) would help readers know what the numeric values in tables represent.

Line 191. I’m not sure the results reveal a “high” level of perceived work-stress and job dissatisfaction, especially when less than half of respondents strongly agreed/agreed. Perhaps that lead-in sentence should be modified to be “Levels of perceived work stress….”

Adding the agree proportions to Table 3 is a good addition. For those items with more “disagree” responses, a kind of “reverse coding” data reporting might be useful.  They could switch to percentage SD/D responses and note that in the footnote, to show how strongly those opinions were.  (The “competent" and “funding” items would be examples; highlight with asterisk perhaps.)  Also, if there might be merit in an “average” score for future comparisons, adding that to the table might be beneficial (and not keeping the authors hamstrung in future studies and comparisons with this effort). Interestingly, for the GHQ-12 items, mean values are reported and used.

Author Response

This revision is an improved version. There are some places that may benefit from additional attention to improve clarity and accuracy.

Thank you. We appreciate that the reviewer believes that the revision was an improved version. We agree and thank the reviewer for their comments.

The revised title did not appear with this version.

We have altered the title to Pharmacists' satisfaction with work and working conditions in New Zealand. An updated survery and a comparison to Canada.

A bit more clarity and explanation on survey administration could benefit others considering this kind of research. Did the NZ Pharmacy Council provide you the email addresses? Or, did they assist by sending an email on your behalf with an invitation and link to the survey? Was the consent based on the general agreement with the Council or did consent specific to this survey occur (as suggested later).

We have added that the NZ Pharmacy Council provided email addresses to us such that we could invite individuals to complete the survey anonymously.  The individuals in the NZ Pharmacy Council's database have provided consent to the Council to supply their contract information to researchers to invite them to participate in studies.  All of the participants provided informed consent to us when they complete the survey.

In the methods, adding the specific labels used for the Likert scaling (e.g., strongly agree, agree, mildly agree, and conversely for disagree) would help readers know what the numeric values in tables represent.

In the Appendix, we have provided the survey that we utilised that clearly has the specific labels for each type of Likert scale that we used.

Line 191. I’m not sure the results reveal a “high” level of perceived work-stress and job dissatisfaction, especially when less than half of respondents strongly agreed/agreed. Perhaps that lead-in sentence should be modified to be “Levels of perceived work stress….”

We have adapted the sentence as per the reviewer's comments.

Adding the agree proportions to Table 3 is a good addition. For those items with more “disagree” responses, a kind of “reverse coding” data reporting might be useful. They could switch to percentage SD/D responses and note that in the footnote, to show how strongly those opinions were. (The “competent" and “funding” items would be examples; highlight with asterisk perhaps.) Also, if there might be merit in an “average” score for future comparisons, adding that to the table might be beneficial (and not keeping the authors hamstrung in future studies and comparisons with this effort). Interestingly, for the GHQ-12 items, mean values are reported and used.

We thank the reviewer for their comments.  We have added the proportion that disagreed/strongly disagreed in a footnote to Table 3 for "Felt unable to remain competent at work", and "Government funding for patient care has increased".  As mentioned previously, we have reported the results the way that we have to facilitate comparisons with the previous survey.